# The effect of temperature on host patch exploitation by an egg parasitoid

**Julie Augustin**[1,2]*, **Guy Boivin**[2], **Gaétan Bourgeois**[2], **Jacques Brodeur**[1]

**1** Département de sciences biologiques, Institut de recherche en biologie végétale, Université de Montréal, Montréal, Québec, Canada, **2** Agriculture and Agri-Food Canada, Saint-Jean-sur-Richelieu Research and Development Centre, Saint-Jean-sur-Richelieu, Québec, Canada

* julie.augustin4@gmail.com

## Abstract

The effect of temperature during host patch exploitation by parasitoids remains poorly understood, despite its importance on female reproductive success. Under laboratory conditions, we explored the behaviour of *Anaphes listronoti*, an egg parasitoid of the carrot weevil, *Listronotus oregonensis*, when foraging on a host patch at five temperatures. Temperature had a strong effect on the female tendency to exploit the patch: *A. listronoti* females parasitized more eggs at intermediate temperature (20 to 30°C) compared to those foraging at the extreme of the range (15.9°C and 32.8°C). However, there was no difference in offspring sex-ratio and clutch size between temperature treatments. Mechanisms of host acceptance within a patch differed between temperatures, especially at 32.8°C where females used ovipositor insertion rather than antennal contact to assess whether a host was already parasitized or not, suggesting that host handling and chemical cues detection were probably constrained at high temperature. Females spent less time on the host patch with increasing temperatures, but temperature had no effect on patch-leaving rules. Our results show that foraging *A. listronoti* females behave better than expected at sub-optimal temperatures, but worse than expected at supra-optimal temperatures. This could impair parasitoid performance under ongoing climate change.

## 1. Introduction

When reproductively active, female parasitoids spend a relatively large proportion of their time searching for and exploiting host patches [1]. In the context of the optimal foraging theory, the marginal value theorem (MVT; [2]) explores searching and oviposition decisions made by parasitoid females. The MVT predicts the duration a female should spend on host patches of various qualities, as modulated by travel duration between patches. Several decision rules and statistical models have been developed to assess the extent to which a female should exploit a patch before leaving [3]. Some of them stem from the idea that, unlike an assumption from MVT, females are not omniscient and need to continuously acquire information from the patch when foraging [4]. Acceptance or rejection of a given host has often an incremental or decremental effect on the propensity to stay or leave a patch, depending on the species

**Funding:** Financial disclosure statement: This work was supported by internal funding from Agriculture

and Agri-Food Canada. GB and GB received the funding. The funders had no role in study design, data collection and analysis, decision to publish, or preparation of the manuscript.

**Competing interests:** The authors declare no competing interests.

studied [3,5]. Several factors may affect patch residence time, including patch quality, female age, egg load, previous experiences and abiotic conditions [3]. However, the role of temperature, a decisive biotic factor influencing behavioural components of ectothermic organisms [6], on host patch exploitation has scarcely been investigated [7], despite the expected more extreme and variable thermic conditions under climate change scenarios [8].

Insect performance is typically maximized along a range of temperatures. Outside this range, performances are reduced until temperature in both the low and high ends reaches a lethal point [9]. However, a change in performance for a trait or behaviour does not necessarily infer that an individual is operating under physiological constraints. In some cases, a behavioural change would be an adaptation to local conditions in order to gain or maintain fitness. For example, a parasitoid female produces more sons at high temperature to maximize her reproductive fitness [10]. High temperature during insect development usually results in individuals developing faster and being smaller at the adult stage than individuals reared at low temperature (temperature size-rule (TSR); [11,12]), with adult size having more fitness consequences for females than for males [13,14]. As shown by Moiroux et al. [10] and further reviewed by Abram et al. [6], it can be difficult to disentangle behaviours that represent an adaptive response to abiotic conditions from those that result from a physiological constraint due to the effect of temperature on metabolic function. Abram et al. [6] suggested the use of a kinetic null temperature model to unravel the behavioural adaptive responses from the constrained physiological responses. Such a null model describes only the kinetic effects of temperature on insect behaviour and allows comparing this response to other potential integrated behaviours. Integrated behaviours result from the assimilation of the thermic information by the central nervous system of the insect, followed by a consequent behavioural response.

Under laboratory conditions, we examined the influence of temperature on host patch exploitation by *Anaphes listronoti* Huber (Hymenoptera: Mymaridae), an egg parasitoid of the carrot weevil *Listronotus oregonensis* LeConte (Coleoptera: Curculionidae). *Anaphes listronoti* females are moderately synovigenic, as they possess 70% of their final egg load at emergence [15], and facultatively gregarious, as 1 to 6 individuals can develop in the same host [16]. Females of this genus can distinguish between host parasitized by themselves, their conspecifics and other species [17]. Exposition to cold temperature during pupal stage can affect their learning capacity [18], the number of eggs laid, their ability to discriminate between parasitized and unparasitized hosts, and patch-leaving rules [19]. However, the effects of adverse temperature during patch exploitation remain to be explored.

The first objective of this study was to examine elements of the host patch exploitation behaviour of *Anaphes listronoti* at sub- and supra-optimal temperatures: residence time and time allocation, patch-leaving rules percent parasitism, offspring sex-ratio and clutch size. According to MVT [2], a female should optimize her time exploiting a patch based on both its quality and transit time between patches. We predicted that females spend less time on host patch as temperature increases, because of two potential phenomena: (i) the increase in metabolic rate with temperature, leading to an acceleration of all behaviours (walking speed, host handling and oviposition), resulting in the patch being exploited faster (kinetic response), and (ii) an increase in subjective duration perceived by the female at higher temperature [20], leading to a decreased residence time (integrated response).

The second objective was to differentiate between physiologically-constrained behaviour and adaptive behaviour of females exploiting a patch at different temperatures using a kinetic null model. If the response of females to temperature is kinetic (i.e. it follows the null model), studied behaviours (number of ovipositions, antennal rejections, ovipositor rejections, residence times) will vary proportionally to females walking speed. Walking speed is a proxy for kinetic response, as it is proportional to metabolic rate [21,22], leading to an increase in

performance as temperature increases until the reach of a tipping point, after which performance decreases [23,24]. Sex allocation and clutch size should however remain unchanged, as well as patch-leaving rules. In the case of an integrated response, we expect the maintenance or an increase of the number of ovipositions along the tested thermal range. We also anticipate an increase in sex ratio at high temperatures, because individuals are smaller when they develop at high temperatures (TSR, [11,12]), and being big is more important for females than for males [13,14]. Accordingly, we expect females to lay a smaller clutch size as temperature increases, because siblings developing from the same host are smaller than those that developed alone in the host [15]. Facultatively gregarious females could compensate for smaller offspring caused by TSR by laying only one egg per host.

In addition, we hypothesized that extreme temperatures would create additional constraints (indirect consequences of metabolic rate), leading to a decrease in efficiency for the following parameters: (i) handling time [25], (ii) egg fertilization (through physical impairment of the spermatheca muscular contraction or sperm motility [10] leading to an increase in primary sex ratio (proportion of males), and (iii) detection of chemical cues [26] used to assess patch quality (e.g. kairomones concentration) [27] or parasitism status (unparasitized *vs* parasitized host) [19]. For females exhibiting one or several of these behaviours, we predicted a reduced performance compared to the kinetic null model. On the other hand, females could still maximize their performance under non-optimal environmental conditions by adapting their behaviour to increase their own fitness or the fitness of their offspring through integrated response to temperature, for example through maternal effect [28]. In this case, we should observe a higher performance than predicted under the null model. We defined performance here as the reproductive success of females, i.e. parasitism rate.

## 2. Material and methods

### 2.1. Parasitoids

*Anaphes listronoti* was reared on carrot weevil eggs in the laboratory at 24˚C ± 2˚C, 50% RH (relative humidity) and 16L:8D following Boivin [29]. The *A. listronoti* strain originated from an Agriculture and Agri-Food Canada (AAFC) untreated carrot field in Sainte-Clotilde, Québec, Canada (45˚09´N, 73˚41´W). Carrot weevil eggs came from a colony established at the AAFC research center in Saint-Jean-sur-Richelieu [30]. Eggs were less than 24 h old when parasitized by *A. listronoti*.

### 2.2. Effect of temperature on patch exploitation

In order to examine the effect of temperature on host-patch exploitation behaviour, females foraging behaviour on host patch was tested at 5 temperatures: 15.9, 20.2, 24.9, 28.4, and 32.8˚C (maximum temperature variation = 1.4˚C; N = 267). During pre-tests, no oviposition occurred at 10˚C (N = 5) and all wasps were dead after 1 h acclimation at 40˚C (N = 5). Temperatures were measured in the experimental arena (Petri dish) with a thermocouple (Omega, model HH23). Host patches consisted of 16 freshly extracted carrot weevil eggs laid by different females from the laboratory colony. Eggs were placed inside a Petri dish (5.6 mm diameter) on a humidified filter paper, along a 1.5 cm square grid, consisting of 4 x 4 eggs. Each egg on the grid was given a position (rows: A, B, C, D and columns: 1, 2, 3, 4) so what emerged of the host egg could be linked to the behaviour of the mother. A central area (3 cm diam) in the open arena was marked with a thin pencil line to delimitate the patch area (See Supplementary material). A one-day-old mated naive (no previous contact with hosts) female contained in a 300 μL Beem® polyethylene capsule and the Petri dish containing the eggs were placed in a growth chamber (Sanyo, model MLR351H) during 1 hour for acclimation to tested

temperatures. The female was next released in the center of the host patch and, based on pre-tests, videotaped for 3 h or until she left the patch for 2 min. At the end of the behavioural observations, eggs were placed individually in identified 300 μL Beem® polyethylene capsule and reared at 24°C ± 2°C, 50% RH, 16L:8D until host or parasitoid emergence. Each female was only tested once. Experiments run between 10:00 and 16:30. Sample sizes were 40, 38, 38, 29 and 40 for 15.9, 20.2, 24.9, 28.4, and 32.8°C, respectively.

### 2.3. Video analyzing

Only videos of females that laid at least one female progeny were included in the analysis (N = 4, 14, 20, 23 and 20 at 15.9, 20.2, 24.9, 28.4, and 32.8°C, respectively). Female that laid only sons were considered unmated and thus removed from analysis (N = 7) because virgin females could behave differently from mated ones [31]. The proportion of females that repro-duced was calculated for each temperature. The frequency and duration of three types of inter-actions with host eggs were quantified: (i) antennal rejection—the female moves away from the host egg following antennal contact, (ii) ovipositor rejection—the female climbs on the host egg, inserts her ovipositor for a short period of time (a few seconds to a few minutes) and then leaves, and (iii) oviposition—the female climbs on the host egg, inserts her ovipositor for several minutes, marks the egg by walking around it and then leaves. We also calculated time spent by the female without interacting with the host (exploration, resting, grooming) and the frequency of exploration bouts outside the host patch. Finally, we calculated residence time on the patch, *i.e.* the time the female spent on the patch before leaving it for more than 2 min.

### 2.4. Offspring production

Parasitoid emergence was checked every 2 to 3 days. At 25°C, development lasts 10 to 11 days [32]. After 21 days, the remaining eggs were dissected and the content identified as either a wasp, a weevil larva or an aborted egg. Wasps detected from dissection were included in the offspring data (N = 77, over 171 dissected eggs). In *A. listronoti*, primary sex ratio cannot be determined during oviposition as for other egg parasitoid species [33]. The sex ratio was there-fore assessed using emerged wasps. Clutch size was also noted.

### 2.5. Null kinetic model

In order to differentiate between kinetic and integrated responses, we compared *A. listronoti* parasitism in relation to temperature to a null kinetic model. Previous data on *A. listronoti* walking speed under different temperatures [34] were used as a proxy for the null thermic model (sensu [6], i.e. the kinetic effect of temperature on behaviour, without further response or constraint from the organism). Walking speed (mm s$^{-1}$) in relation to temperature was cal-culated using the following equation:

$$\text{Walking Speed (T)} = a * \text{T} + b \tag{1}$$

where T is the temperature (°C), *a* represents the increase in walking speed corresponding to an increase in temperature, with a value of 0.01184, and *b* represents the theoretical walking speed of *A. listronoti* females at 0°C, with a value of 0.65471.

### 2.6. Data analyses

Linear and polynomial regressions were used to analyse the effect of temperature on the differ-ent variables studied (oviposition/rejection behaviours and offspring parameters). The degree of polynomial regression chosen was the lowest polynomial degree having significant

polynomial coefficients ($P<0.05$) and the highest $R^2$ value. A Mann-Whitney test was used to compare the number of observed ovipositions to the number of eggs from which a parasitoid wasp had emerged (realised ovipositions). A Kruskal-Wallis test was used to determine the effect of temperature on the percentage of aborted eggs, and Wilcoxon tests were used to analyse differences between treatments. A Cox's proportional hazards model [35] was used to identify cues used by females to leave the patch as related to temperature. The Cox model is expressed in term of hazard rate or the probability per unit of time that a female would leave the patch. It is assumed that the hazard rate is the product of a baseline hazard and the effects of all the explanatory variables. The model is expressed as:

$$h(t;z) = h_0(t)\exp\left\{\sum_{i=1}^{p} \beta_i z_i(t)\right\} \tag{2}$$

where $h\,(t;\,z)$ is the hazard rate, $h_o\,(t)$ the baseline hazard, $t$ the time since the female entered the patch, $\beta_i$ the regression coefficient giving the relative contribution of the covariates $z_i\,(t)$. The hazard ratio, given by the expression $exp(\beta_i\,z_i)$, determines whether the covariable $z_i$ influences the patch-leaving tendency of the females. If the hazard ratio is inferior to one, the female patch-leaving tendency is reduced by this covariable. If the hazard ratio is higher than one, the patch-leaving tendency is increased by the covariable [5,19]. The variables tested were: number of antennal and ovipositor rejections, number of ovipositions, number of times the female exited the patch (for less than 2 min), number of rejections of non-parasitized hosts, number of rejections of parasitized hosts, rates of oviposition and rejection. The effect of each variable and the overall significance of the model were tested using likelihood ratio tests.

## 3. Results

### 3.1. Effect of temperature on host-patch exploitation behaviour

**3.1.1. Patch acceptance.** Fewer females *A. listronoti* exploited the patches at low and high temperatures than at intermediate temperatures (second degree polynomial, $p \leq 0.001$, $R^2 = 0.16$; Fig 1), with patch acceptance being maximum at 28.4˚C. The total number of females accepting the patch was low at 15.9˚C (N = 4), and analyses were run with and without them. There was no difference in the significance of tested variables with and without the 15.9˚C treatment, except for duration of ovipositor rejection and clutch size, with lost and gained significance without the 15.9˚C treatment, respectively. Therefore, we kept the 15.9˚C treatment in the analyses.

**3.1.2. Patch time allocation.** Patch time allocation was significantly affected by temperature (Fig 2a), with differences in both the frequency and duration of tested behaviours. Temperature had a significant effect on the number of antennal rejections (p = 0.004), ovipositor rejections (p = 0.005) and ovipositions ($p \leq 0.001$; Fig 2b). Females spent overall more time performing antennal rejection at low and intermediate temperatures compared to 32.8˚C ($p \leq 0.001$, $R^2 = 0.29$; Fig 2a), due to a decrease in both frequency and duration at high temperature ($p \leq 0.001$; Fig 2b). In contrast, they spent more time performing ovipositor rejections at 32.8˚C compared to other temperatures ($p \leq 0.001$, $R^2 = 0.31$; Fig 2a), due mainly to a higher frequency of this behaviour. The durations of ovipositor rejections and ovipositions decreased from 15˚C to 28.4˚C but increased at 32.8˚C (p = 0.002 and $p \leq 0.001$, respectively). Despite this change in oviposition duration, females spent relatively more time laying eggs at intermediate temperatures than at low and high temperatures (p = 0.01, $R^2 = 0.09$; Fig 2a) because of the increased frequency of the behaviour. There were no significant relationships between

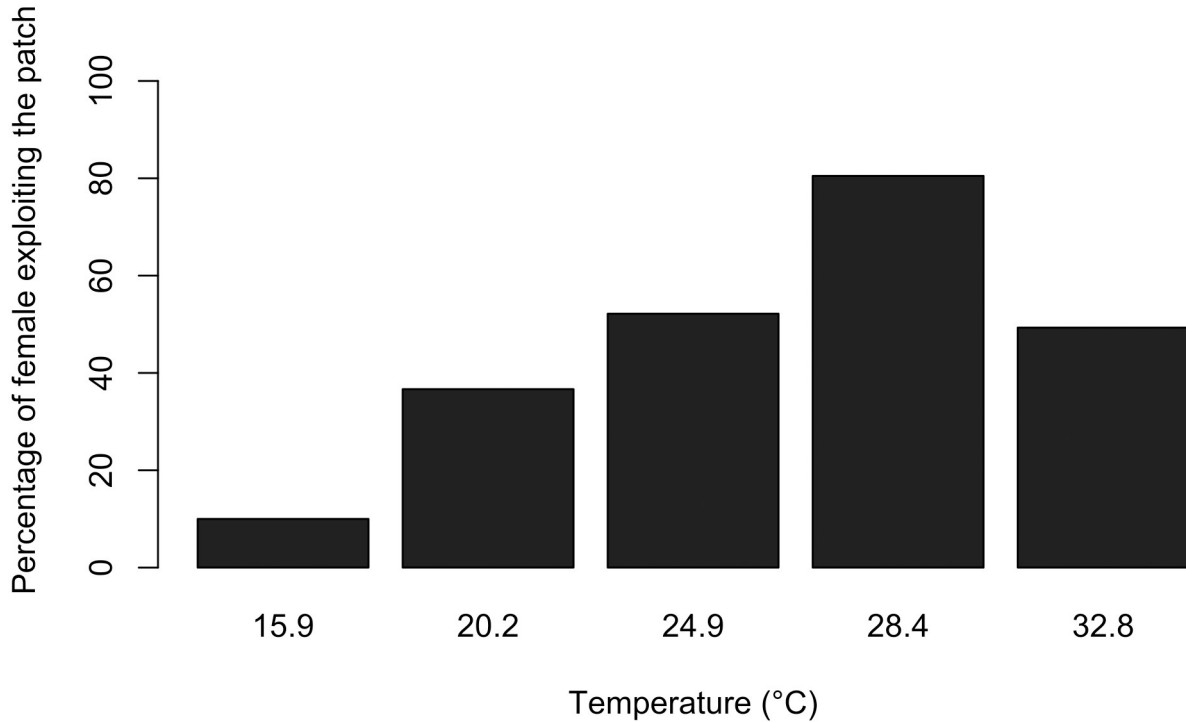

**Fig 1. Percentage of females *Anaphes listronoti* that oviposited in at least one host egg (*Listronotus oregonensis*) at different temperatures.**

temperature and the number and duration of other behaviours (exploring, resting and grooming, p = 0.09) or the frequency of exploration outside the host patch (p = 0.14).

**3.1.3. Patch-exploitation strategy.** Females left the host patch earlier as temperature increased (p = 0.008; Fig 3), but cues used in patch-leaving decisions did not change with temperature (Table 1). Females' tendency to leave the patch increased together with oviposition and rejection rates at almost all temperatures. All other tested covariables had no effect. There was a significant correlation between the number of rejections and the number of oviposition (Pearson's correlation: 0.62 for antennal rejection and 0.24 for ovipositor rejection; p ≤ 0.001 et p = 0.03, respectively).

*Offspring number and sex-ratio*. Females parasitized fewer eggs at high and low temperatures (2nd degree polynomial regression, p ≤ 0.001, $R^2$ = 0.50) than at intermediate temperatures (Fig 4). Percent parasitism was higher when estimated from video sequences than when calculated from parasitoid emergence and host egg dissection (Mann-Whitney test, p = 0.009): 13.6 ± 16.9% of observed parasitized eggs yielded no offspring. This effect was greater at 32.8˚C (29.7 ± 23.6%) than at 20.2˚C, 24.9˚C and 30˚C (p = 0.03, p = 0.001 and p ≤ 0.001, Fig 5). Temperature had no significant effect on offspring sex-ratio (linear regression: p = 0.35, mean sex-ratio: 0.16 ± 0.11) and clutch size (linear regression: p = 0.08, mean clutch size: 1.26 ± 0.22). However, when the 15.9˚C treatment was removed from analysis, clutch size significantly decreased with temperature (linear regression: p = 0.02).

### 3.2. Null kinetic model

*Anaphes listronoti* performance, expressed by percent parasitism, was maximum at 28.4˚C (Fig 5). It was higher than expected under the null kinetic model at all sub-optimal temperatures

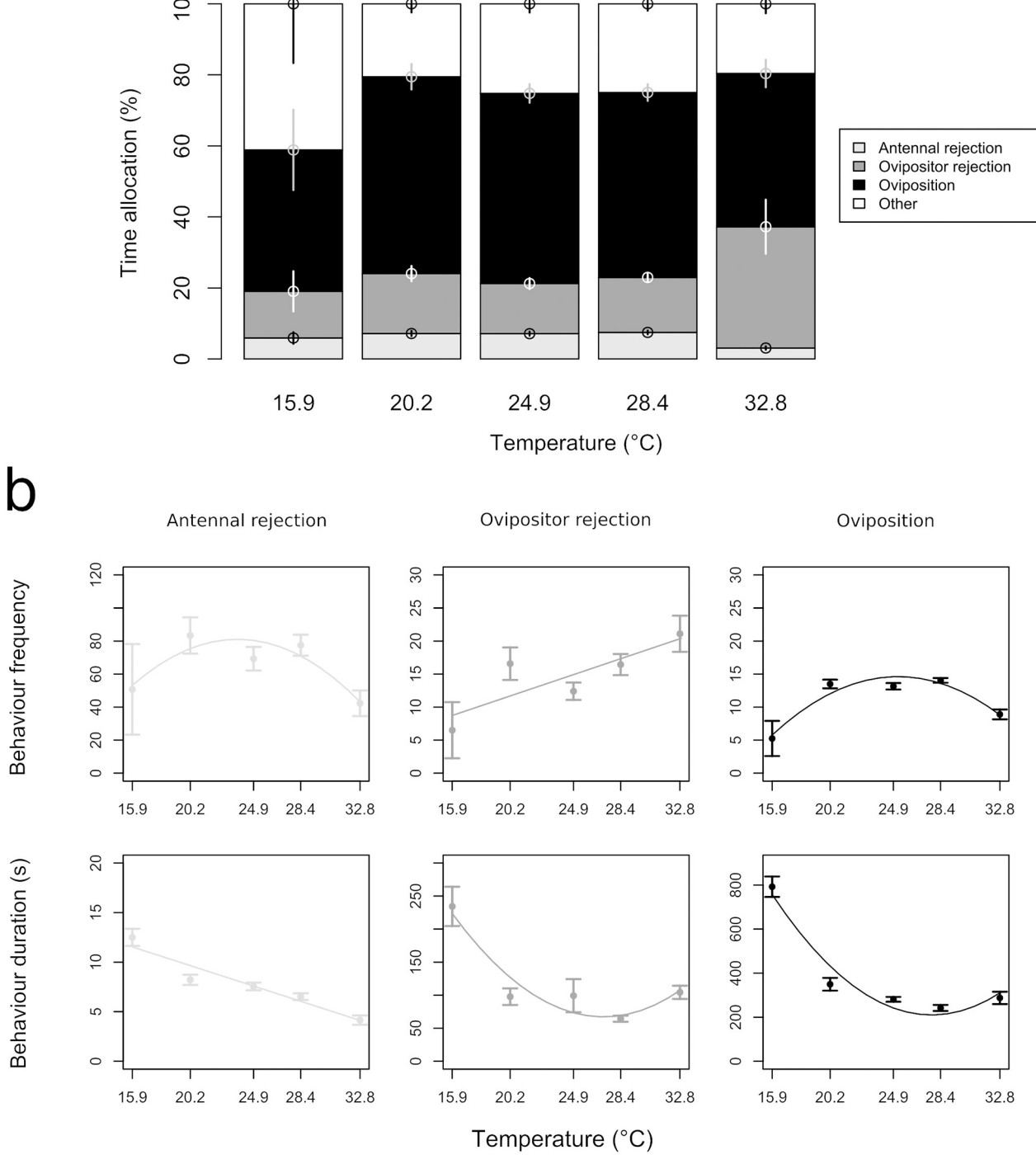

**Fig 2.** (a) Time allocation of behaviours of *Anaphes listronoti* females during host patch exploitation (*Listronotus oregonensis*) as related to temperature. Other behaviours include grooming, resting and walking. ($\bar{\text{X}} \pm \text{SE}$) (b) Frequency and duration of *A. listronoti* females' behaviours at different temperatures.

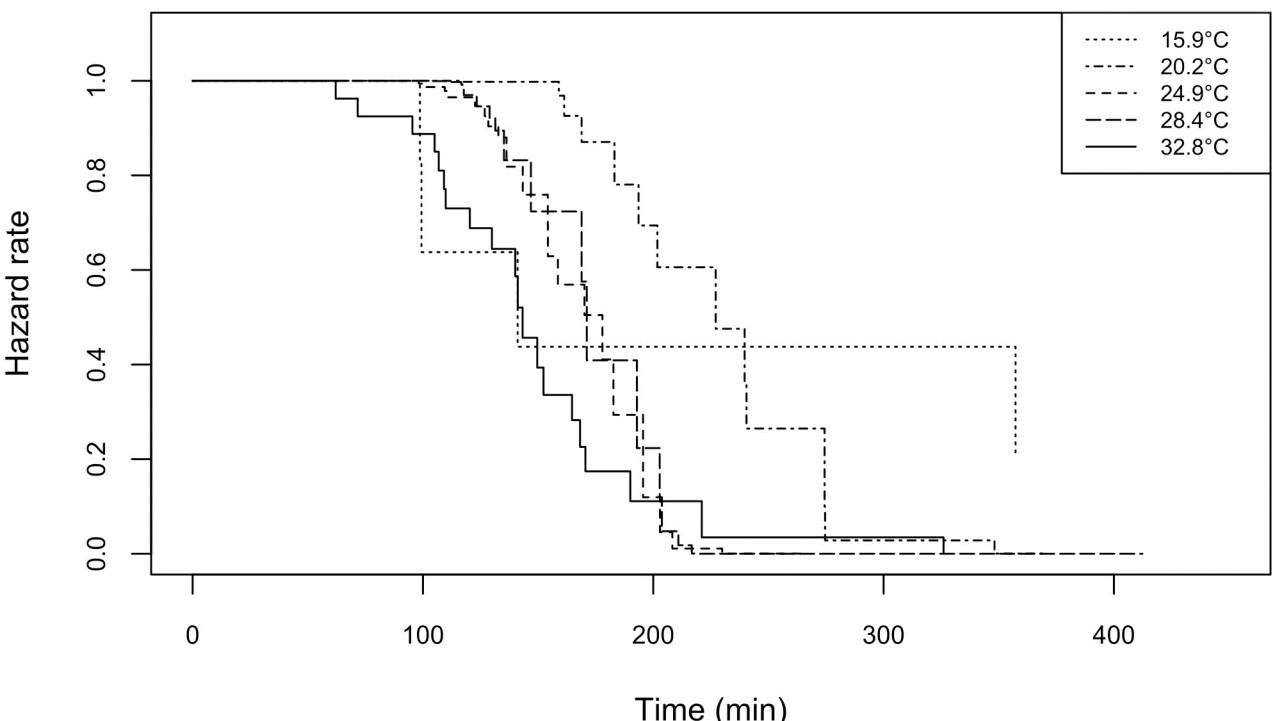

**Fig 3. Patch-leaving hazard rate of *Anaphes listronoti* females when exploiting *Listronotus oregonensis* egg masses at different temperatures.**

and lower than expected at the supra-optimal temperature. Duration of antennal rejection and patch residence time were the only variables proportional to walking speed (S1 Fig). For other variables, most significant responses followed a bell curve or a reverse bell curve along the thermic range and thus differed from the kinetic null model (S1 Fig).

## 4. Discussion

Temperature significantly influenced host patch exploitation by *A. listronoti* females under laboratory conditions, with maximum parasitism observed at 28.4˚C. They spent less time on the patch with increasing temperatures and parasitism rate was reduced at low and high temperatures, but there was no difference in offspring sex-ratio or clutch size. Time allocation during patch exploitation differed between temperatures, especially at 32.8˚C where females switched from antennal to ovipositor rejections while using the same cues for patch leaving decisions, i.e. oviposition and rejection rate. The null model showed that *A. listronoti* performance was higher than expected under kinetic constraint only for sub-optimal temperatures, but lower than expected at supra-optimal temperatures.

### 4.1. Effect of temperature on host-patch exploitation behaviour

**4.1.1. Patch acceptance.** At all tested temperatures, except 28.4˚C, less than half of the females exploited the host patch (Fig 1), probably resulting from the experimental conditions. *Anaphes listronoti* tendency to leave the host patch without exploiting it was significantly affected by temperature. In parasitoids, warm temperatures are usually associated with higher dispersal through increase in walking and flying capacities [36–39]. However, here we report the highest patch departure rates at the highest and the lowest temperatures tested. In an aphid-ladybeetle system, Sentis et al. [26] showed that temperature modified the capacity of

Table 1. Effects of different behavioural components on residence time of *Anaphes listronoti* females at different temperatures when exploiting *Listronotus oregonensis* egg patches.

| | 15.9°C | | | 20.2°C | | | 24.9°C | | | 28.4°C | | | 32.8°C | | |
|---|---|---|---|---|---|---|---|---|---|---|---|---|---|---|---|
| | β ± se | exp(β) | p | β ± se | exp(β) | p | β ± se | exp(β) | p | β ± se | exp(β) | p | β ± se | exp(β) | p |
| Number of antennal rejections | -0.01 ± 0.02 | 0.99 | 0.4 | 0.02 ± 0.01 | 1.02 | 0.1 | 0.006 ± 0.008 | 1.006 | 0.5 | -0.005 ± 0.007 | 0.99 | 0.5 | -0.003 ± 0.009 | 0.99 | 0.7 |
| Number of ovipositor rejections | -0.09 ± 0.13 | 0.91 | 0.4 | -0.005 ± 0.04 | 0.99 | 0.9 | 0.04 ± 0.04 | 1.04 | 0.3 | -0.05 ± 0.04 | 0.95 | 0.2 | -0.02 ± 0.02 | 0.98 | 0.4 |
| Number of ovipositions | -0.14 ± 0.2 | 0.87 | 0.4 | -0.07 ± 0.11 | 0.93 | 0.5 | 0.07 ± 0.09 | 1.08 | 0.4 | -0.03 ± 0.14 | 0.98 | 0.9 | -0.09 ± 0.08 | 0.92 | 0.3 |
| Rejections of unparasitized host | 0.33 ± 0.26 | 1.4 | 0.05 | 0.01 ± 0.01 | 1.01 | 0.4 | 0.04 ± 0.02 | 1.04 | 0.09 | -0.003 ± 0.02 | 0.99 | 0.9 | -0.02 ± 0.01 | 0.98 | 0.2 |
| Rejections of parasitized host | -0.02 ± 0.02 | 0.98 | 0.3 | 0.01 ± 0.01 | 1.01 | 0.3 | 0.003 ± 0.008 | 1.003 | 0.7 | -0.007 ± 0.007 | 0.99 | 0.3 | 0.004 ± 0.008 | 1.004 | 0.6 |
| Total rejections | -0.01 ± 0.02 | 0.99 | 0.4 | 0.01 ± 0.01 | 1.01 | 0.2 | 0.006 ± 0.007 | 1.006 | 0.4 | -0.005 ± 0.006 | 0.99 | 0.4 | -0.004 ± 0.008 | 0.99 | 0.6 |
| Number of patch exits | -0.5 ± 0.49 | 0.61 | 0.09 | 0.006 ± 0.03 | 1.006 | 0.8 | 0.07 ± 0.03 | 1.07 | 0.04 * | -0.03 ± 0.02 | 0.97 | 0.06 | -0.02 ± 0.05 | 0.98 | 0.6 |
| Oviposition rate | 27.2 ± 45 | $6.63 \times 10^{11}$ | 0.5 | 83.1 ± 26.1 | $1.2 \times 10^{36}$ | ≤ 0.001 *** | 82.9 ± 18.5 | $1.09 \times 10^{36}$ | ≤ 0.001 *** | 103.5 ± 23 | $9.08 \times 10^{44}$ | ≤ 0.001 *** | 23.9 ± 10.7 | $2.53 \times 10^{10}$ | 0.02 * |
| Rejection rate | 1.45 ± 3.6 | 4.25 | 0.7 | 4.71 ± 1.55 | 110.5 | ≤ 0.001 *** | 3.11 ± 1.07 | 22.51 | 0.004 ** | 3.68 ± 1.2 | 39.46 | 0.001 *** | 1.13 ± 0.72 | 3.1 | 0.1 |

Estimated regression coefficients β ± se, hazard ratios exp(β), and p-values were calculated from Cox regression models and likelihood ratio tests, respectively.

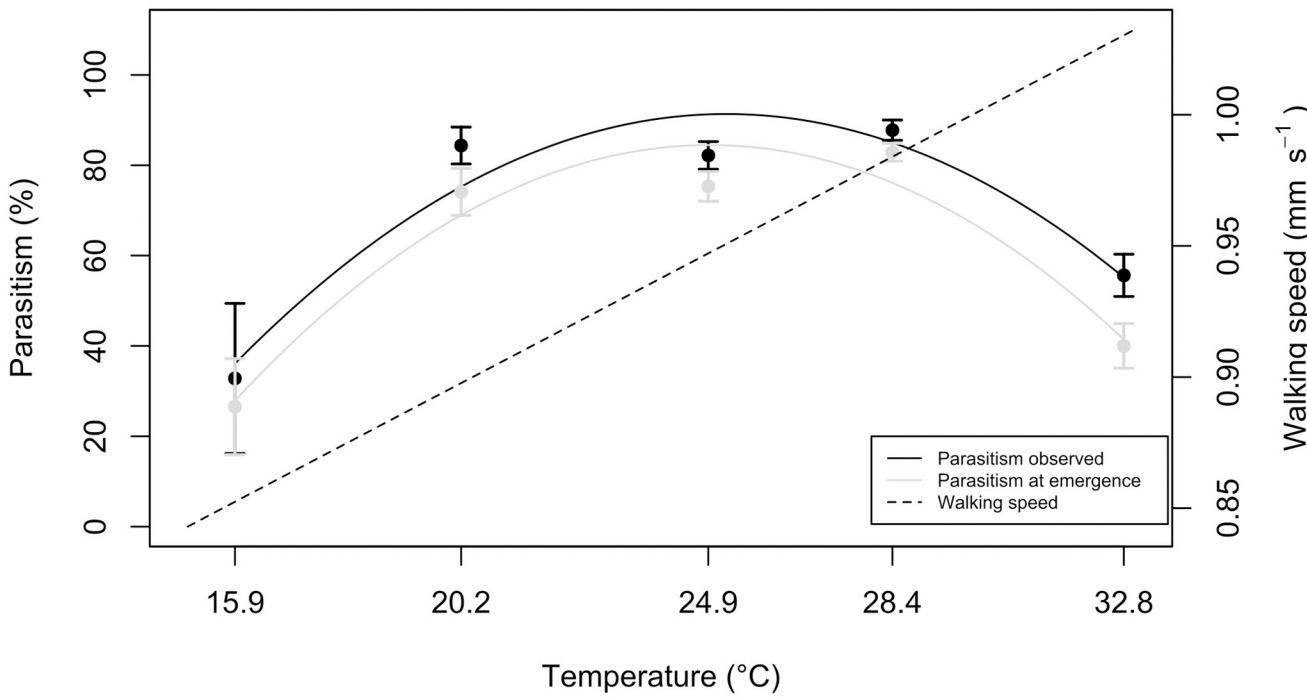

**Fig 4. Percentage of *Anaphes listronoti* parasitism of *Listronotus oregonensis* eggs ($\bar{X} \pm$ SE), calculated from either video sequences (black) or offspring emergence (grey), and walking speed calculated from Eq 1 (see text).**

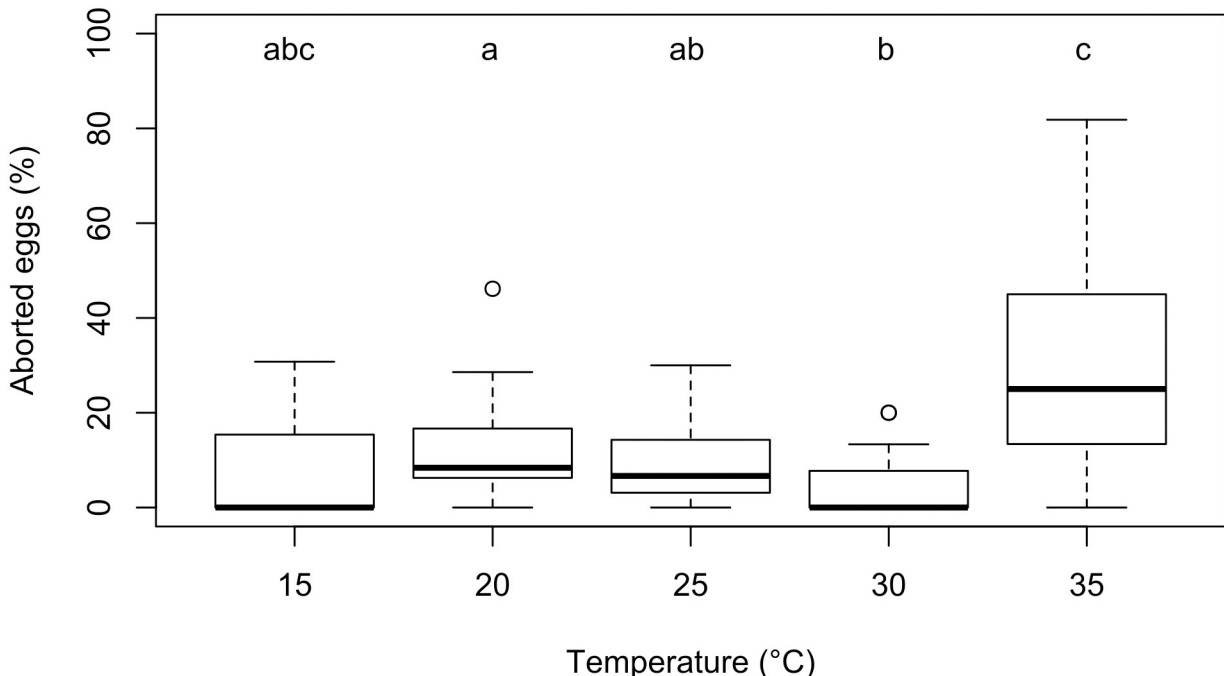

**Fig 5. Percentage of *Listronotus oregonensis* aborted eggs in patches exploited by *Anaphes listronoti* females at different temperatures.**
Boxes represent the 25th and 75th percentiles, the heavy line represents the median, whiskers represent 1.5 × interquartile range of the data, and data points represent outliers. Different letters represent significant differences between temperatures.

the predator to perceive the nature and quantity of infochemicals emitted by the prey. Thus, high levels of *A. listronoti* patch departure at lowest and highest temperatures may result from a lack of kairomones detection, or their misreading by searching females.

**4.1.2. Patch time allocation.** The duration of antennal rejections closely followed the kinetic response, as inferred by walking speed; the higher the temperature, the faster the metabolism due to protein conformation and binding to substrate [40]. However, durations of ovipositor rejection and oviposition increased at 32.8˚C, suggesting a constraint to host handling at high temperature. Similarly, Sentis et al. [25] found that handling rate in lady-beetles were lower than expected under the metabolic theory of ecology (MTE) [41] at high temperatures.

*Anaphes listronoti* females spent more time completing ovipositor rejection and less time performing antennal rejection at 32.8˚C. Females mark the host eggs after oviposition by touching its surface with the antennae while walking over it [33]. They can learn to associate these external marking cues, detected with antennae, with the presence of a parasitoid egg inside the host [42]. Yet, after exposure to cold at the pupal stage, they are less efficient in assessing host status with external examination only, and need to insert their ovipositor in the host to assess whether it was already parasitized [19]. In general, thermal stress affects both memory consolidation and the rate of the active forgetting process ([43] and references therein). However, in our case, it is not clear whether thermal stress impaired the detection or learning of chemical cues. The observed changes in behaviour could arise from: (i) alterations in the female's host marking behaviour right after oviposition, (ii) changes in the nature and/ or quantity of marking pheromones as a result of temperature, occurring between host mark-ing and the next encounter, (iii) changes in the female's capacity to recognize kairomones, (iv) a combination of some or all of the above. In another study, we found that *A. listronoti* males failed to exhibit courtship, and females refused to mate as frequently at low and high tempera-tures than at the optimum [44]. This could also result from thermal-induced impairment of the capacity of females to detect/recognize chemical cues.

**4.1.3. Patch exploitation strategy.** *Anaphes listronoti* residence time decreased with tem-perature, suggesting that host patch exploitation is, at least partially, linked to the metabolic rate of the individual. Females used the same patch-leaving rules (i.e. oviposition rates) at all temperatures, except at 15.9˚C where we identified no behavioural components affecting patch leaving rate of females. Rejection rates also significantly increased patch-leaving at 20.2, 24.9, and 28.4˚C, but not at 32.8˚C. Number of rejections and ovipositions are correlated, sug-gesting that they are both used as patch-leaving rules at almost all temperatures. At the highest temperature, females may have had difficulties assessing whether the host was parasitized or not (see above).

Parasitoid females entering a host patch have an initially tendency to remain on it [27]. This propensity decreases with time until it reaches a threshold, at which females leave the patch. When a host encounter, a rejection or an oviposition increase patch residence time, the underlying mechanism is defined as incremental. On the opposite, when these interactions with the host increase patch-leaving rates, the mechanism is decremental [27]. *Anaphes listro-noti* females thus operate under a decremental mechanism, because ovipositions and host rejections increased patch leaving. Decremental mechanisms are considered optimal for spe-cies whose host patches are homogeneous in quality through the environment [3,45]. Under these conditions, females could rely on the same patch-leaving rules, as we observed in this study, despite changes in temperature conditions. This could allow females performance at low temperature to be higher than expected under the null kinetic model, as seen in Fig 4. However, at the hottest temperature, performance was lower than expected. Females likely lost time when assessing host quality because they performed ovipositor rejection, that takes on

average 14.1 times longer than antennal rejection, to discriminate unparasitized *vs* parasitized hosts.

Changes in residence time in relation to temperature followed changes in walking speed, but they were not directly proportional. A two-fold increase in walking speed resulted in a one-fold decrease in residence time, suggesting that metabolic rate is not the only mechanism responsible for residence time in *A. listronoti*. We hypothesized that females spend less time on the patch as temperature increases, and this effect should be stronger than metabolic rate alone in the case of an integrated response. However, we observed something different: the magnitude of changes in residence time is less than those in walking speed. Using the same patch-leaving rules could create a buffer effect that homogenize residence time even when female parasitoids experience different conditions. This could also be explained by the alteration of the female time perception by temperature: for ectotherms, time subjectively passing by faster at high than at low temperature [20]. Thus, at high temperature, from a female's perspective, relatively more time was spent on the patch. Accordingly, a female's perception time would affect her oviposition and rejection rate. In addition, despite the 1-hour acclimation before females were released in the patch, it remains possible that observed behaviours are not a response to the experienced temperature during patch exploitation, but rather a response to the increase or decrease in temperature compared to the pre-trial temperature [46]. Indeed, it could be advantageous for females to adjust patch exploitation (residence time and number of ovipositions) after experiencing a drop in environmental temperature, that could indicate adverse environmental conditions such as rain or wind. In all cases, females spent less time on the patch as temperature increased, and this effect cannot be explained by the kinetic response alone. Unfortunately, our data do not allow a definite identification of the mechanisms or strategies involved.

**4.1.4. Offspring number and sex ratio.** Females laid fewer offspring at low and high temperatures. This may result from reduced activity [37], differences in searching behaviour [47] or the search of thermal refuges instead of hosts. Clutch size ($1.26 \pm 0.22$ eggs per host) is similar to other report for *A. listronoti* at 23°C ($1.29 \pm 0.47$ [16]). Similarly, temperature did not affect clutch size of the egg parasitoid *Trichogramma dendrolimi* [48]. However, when removing 15.9°C from analysis, clutch size significantly decreased with increasing temperatures. Individual size decreases with both high temperatures during development (TSR, [11,12]), and number of siblings that developed in the same host [15]. Size being linked to fitness [13–15] (but see [49,50]), females could decrease their clutch size at high temperatures to compensate for the expected size diminution of their offspring caused by temperature. This could allow them to secure a certain size, and therefore fitness, for their offspring. This hypothesis remains to be tested.

There was a small incidence of parasitoid-induced host egg abortion (*sensu* [51]) from 15.9 to 28.4°C, with parasitism leading to failed parasitoid offspring development and host egg abortion. Percentage of egg abortion was significantly higher at 32.8°C (29.7% of eggs aborted) compared to other temperatures (Fig 5). Egg abortion were possibly caused by the females' several attempts to oviposit before succeeding at high temperatures.

Interestingly, offspring sex ratio did not vary with temperatures experienced by *A. listronoti* females when exploiting a patch. In contrast, Moiroux et al. [10] showed that *T. euproctidis* offspring sex ratio increases at both low and high temperatures experienced by foraging females. Female parasitoids allocate a sex ratio that increases their own fitness or their offspring fitness [28]. For example, they produce more males at high temperature because size decreases with increasing temperatures during development [11,12], and being large is more important for parasitoid female fitness than for male fitness [13,14]. Sex allocation by females can be constrained by defect egg fertilization [10] or sperm depletion [52]. Sex-specific survival [53,54] or dispersal capacity [55] under adverse temperatures can also affect secondary sex ratio. No difference in secondary sex ratio was observed in *A. listronoti*, either because females did not

adjust sex allocation under the range of tested temperatures and there was no difference in mortality between sexes during parasitoid development, or because females did change sex allocation but differential embryonic or larval mortality between sexes resulted in a similar secondary sex ratio at all temperatures. As the duration of the tests only differed by about three hours between thermal treatments, and subsequent development occurred under the same rearing conditions for about 12 days, we expect sex-specific mortality rate to be the same for all treatments. Therefore, we privilege the first explanation, i.e. no differential sex allocation related to temperature. This explanation also fit with the maintenance of the same patch exploitation behaviour: females would not modify their exploitation strategy no matter the ambient temperature.

## 4.2. Null kinetic model

The responses of *A. listronoti* females to temperature during patch exploitation are partially driven by kinetic response. An absolute kinetic response would result in females behaving the same at all temperatures, but slower or faster as temperatures decreased or increased. This is not the case for *A. listronoti* because females had fewer host contacts at low and high temperatures than at intermediate temperatures, and patch exploitation strategies differed between temperatures. As parasitoid performance was lower than expected at high temperatures, we suggest three potential constraints: increased handling time due to impaired coordinated movement [25], weakening of the muscular contraction of the spermatheca, leading to lower egg fertilization [10] and reduced detection of chemical cues [26]. The first and the third constraints were obvious at 32.8°C, with females spending more time interacting with host eggs than expected, rejection rate not being used as a patch leaving rule, and females switching from antennal rejections to ovipositor rejections for host quality assessment. The second constraint, the impairment of the muscular contraction of the spermatheca during oviposition, was not observed; sex ratio remaining constant along the tested thermic range. At low temperatures, performance was higher than expected under the kinetic null model. This could either be an adaptive response of females to temperature or, as suggested earlier, a buffering effect due to use of the same patch-leaving rules.

## 4.3. Conclusion and perspectives

Our results highlight the complexity of parasitoid behavioural responses to temperature. The frequency and duration of behaviours may (i) simply result from changes in metabolic rate due to the kinetic effect of temperature on insects, (ii) be indirect consequences of the insect's metabolic rate, such as handling time, or (iii) result from constraints arising from other sources, such as kairomones emission and perception at sub-optimal temperatures [19,26]. *Anaphes listronoti* females behave better than expected at sub-optimal temperatures, but worse than expected at supra-optimal temperatures. This could have serious consequences under ongoing climate change for such a short-lived species (2 to 3 days at 29°C [16]). Females would have less time to find and exploit hosts, and offspring will be smaller following the temperature size-rule [11,12], and thus have a lower fitness [14]. In contrast, these results suggest that populations from temperate climate may perform better than predicted under the kinetic null model at low temperatures. This is an interesting hypothesis to test in future research.

## Supporting information

**S1 Fig. Supplementary graphics.**
(TIF)

**S2 Fig. Supplementary material—Experimental setup.**
(TIF)

**S1 Table. Regressions statistics.**
(XLSX)

**S1 Data. Raw data behaviour.**
(XLSX)

**S2 Data. Raw data offspring.**
(XLSX)

## Acknowledgments

We thank D. Thibodeau, J. Frenette and J. Vaillancourt for technical assistance. We thank F. Dubois, M. Goubault and J. Moiroux for comments on a previous version of the manuscript, that allowed to significantly improve its quality. We thank E. Duquette for linguistic revision.

## Author Contributions

**Conceptualization:** Julie Augustin, Guy Boivin, Gaétan Bourgeois, Jacques Brodeur.

**Data curation:** Julie Augustin.

**Formal analysis:** Julie Augustin.

**Funding acquisition:** Guy Boivin, Gaétan Bourgeois.

**Investigation:** Julie Augustin.

**Methodology:** Julie Augustin, Guy Boivin.

**Project administration:** Guy Boivin, Gaétan Bourgeois.

**Resources:** Guy Boivin, Gaétan Bourgeois.

**Software:** Julie Augustin.

**Supervision:** Guy Boivin, Gaétan Bourgeois, Jacques Brodeur.

**Validation:** Guy Boivin, Gaétan Bourgeois, Jacques Brodeur.

**Visualization:** Julie Augustin.

**Writing – original draft:** Julie Augustin.

**Writing – review & editing:** Guy Boivin, Gaétan Bourgeois, Jacques Brodeur.

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
