## [Decision Letter · Decision Letter 0]

20 May 2021

PONE-D-21-08181

The effect of temperature on host patch exploitation by an egg parasitoid

PLOS ONE

Dear Dr.Julie Augustin

Thank you for submitting your manuscript to PLOS ONE. After careful consideration, we feel that it has merit but does not fully meet PLOS ONE’s publication criteria as it currently stands. Therefore, we invite you to submit a revised version of the manuscript that addresses the points raised during the review process.

We look forward to receiving your revised manuscript.

Kind regards,

Kleber Del-Claro, PhD

Academic Editor

PLOS ONE

Journal Requirements:

Additional Editor Comments:

Considering two expert reviewers, we could accept your manuscript (please condier all the suggestions!) for publication after minor revision. Congratulations.

Reviewers' comments:

Reviewer's Responses to Questions

**Comments to the Author**

1. Is the manuscript technically sound, and do the data support the conclusions?

Reviewer #1: Yes

Reviewer #2: Yes

2. Has the statistical analysis been performed appropriately and rigorously? 

Reviewer #1: Yes

Reviewer #2: Yes

3. Have the authors made all data underlying the findings in their manuscript fully available?

Reviewer #1: Yes

Reviewer #2: Yes

4. Is the manuscript presented in an intelligible fashion and written in standard English?

Reviewer #1: Yes

Reviewer #2: Yes

5. Review Comments to the Author

Reviewer #1: The authors of MS "The effect of temperature on host patch exploitation by an egg parasitoid" studied how five different temperatures affect the behavior and performance of a parasitoid wasp. They had built a good text and designed an interesting model of study. The introduction is well structured, M&M is replicable, and results and discussion support their proposal. However, the abstract still needs a conclusion. In addition, in the discussion (4.2) the authors pointed "Anaphes listronoti females likely experienced difficulties handling host eggs...". This sounds like an anthropomorphization and I suggest rephrase.

Reviewer #2: Dear authors,

I enjoyed reading the MS and I learned a lot about how temperature affects parasitoids. I made several suggestions and I hope I could enhance the quality of the MS. The text is well written, the figures are good and informative. Since you have many objctives and suggesting improving the presentation of them in Methods, Results and Discussion to a better understanding.

Comments:

Abstract

L 18 – remove “high quality”. Readers must wonder why you did not compare high- and low-quality hosts

I missed a conclusion in the Abstract.

Keywords: “temperature” is already in the title

Introduction

The last paragraph should come before the two objectives

The objectives and hypotheses are described in detail, and this clarifies more the study.

Methods

I suggest indicating which objectives and hypotheses are related to each method. This helps a lot the understanding. I would also recommend this for the Results and Discussion, but authors should think if adding more subsections is appropriate. I think that since your study has many objectives, this might improve the comprehension of the study in its full magnitude. I confess that I felt a bit lost sometimes. For instance, one of you predictions is “(i) an increase in metabolic rate with temperature, leading to an acceleration of all behaviours”. So you could write “in order to test the prediction (i) females were analyzed under five temperatures ….”

Patch exploitation experiment

L. 114 – 117 When were eggs collected? Did eggs come from a single individual plant? Did all eggs come from clone plants?

L 120 – 121 What was the sample size of females in each temperature? I see that this information is shown in Figure 1, but I would like to see it in the text was well

L. 121 – 122 How many observations were performed?

L 123 – 124 Were these weevil eggs from a single female?

L 125 – 127 Is it possible to provide a figure of this experiment? (maybe as online resource)

Video analyzing

L 136 – 138 It is not clear to me why you only analyzed females that laid female progeny

Offspring production

Ok

Data analyses

I am not familiar with the A Cox’s proportional hazards model, so I will assume authors made it right

Results

Looking at figure 2a I do not see much difference among classes, but your analyses show significant results. Have you used the correct statistical tests?

I suggest double checking the tests used. I found a Mann Whitney that is not described in the Methods.

Discussion

All the topics shown in the Results are discussed. Nonetheless, the discussion is too long, more than 2000 words. It is difficult to keep the attention and focus with such a long discussion. Authors should speculate less and concentrate on their own results. A shorter and more objective discussion will be more valuable.

Figures

Very good! Mines are low resolution, but I think that the final version will have figures with high resolution.

6. PLOS authors have the option to publish the peer review history of their article (what does this mean?). If published, this will include your full peer review and any attached files.

Reviewer #1: **Yes: **Dr. Bruno de Sousa-Lopes

Reviewer #2: No

---

## [Author Response · Author response to Decision Letter 0]

16 Jun 2021

Reviewer #1: The authors of MS "The effect of temperature on host patch exploitation by an egg parasitoid" studied how five different temperatures affect the behavior and performance of a parasitoid wasp. They had built a good text and designed an interesting model of study. The introduction is well structured, M&M is replicable, and results and discussion support their proposal. However, the abstract still needs a conclusion. In addition, in the discussion (4.2) the authors pointed "Anaphes listronoti females likely experienced difficulties handling host eggs...". This sounds like an anthropomorphization and I suggest rephrase.

A short conclusion has been added to the abstract - lines 27-28.

The sentence “Anaphes listronoti females likely experienced difficulties handling host eggs..." has been removed because the idea it conveyed was discussed in lines 293-294 and 296-298. 

 

Reviewer #2: Dear authors,

I enjoyed reading the MS and I learned a lot about how temperature affects parasitoids. I made several suggestions and I hope I could enhance the quality of the MS. The text is well written, the figures are good and informative. Since you have many objctives and suggesting improving the presentation of them in Methods, Results and Discussion to a better understanding.

Comments:

Abstract

L 18 – remove “high quality”. Readers must wonder why you did not compare high- and low-quality hosts

I missed a conclusion in the Abstract

Keywords: “temperature” is already in the title

“high quality” has been removed in line 18

A short conclusion has been added - lines 27-28

 “temperature” has been removed from Keywords

Introduction

The last paragraph should come before the two objectives

The objectives and hypotheses are described in detail, and this clarifies more the study.

The last paragraph has been moved before the two objectives. 

Methods

I suggest indicating which objectives and hypotheses are related to each method. This helps a lot the understanding. I would also recommend this for the Results and Discussion, but authors should think if adding more subsections is appropriate. I think that since your study has many objectives, this might improve the comprehension of the study in its full magnitude. I confess that I felt a bit lost sometimes. For instance, one of you predictions is “(i) an increase in metabolic rate with temperature, leading to an acceleration of all behaviours”. So you could write “in order to test the prediction (i) females were analyzed under five temperatures ….”

As suggested by reviewer#2, the text has been modified in places to clarify which experiments or results are linked to each objectives: 

- Presentation of objectives and hypotheses have been modified lines 79-81,

- Explanatory sentences have been added in lines 118, 157-158,

- Titles of sections have been modified in Material and methods in line 117. Subsections have been added or modified in the Results in lines 191-192, 203, 222, 237, 251 and in the Discussion in lines 276-277, 287, 308, 347, 380, 397.

Patch exploitation experiment

L. 114 – 117 When were eggs collected? Did eggs come from a single individual plant? Did all eggs come from clone plants?

Eggs were collected from the carrot weevil colony, and were less than 24h old when given to the parasitoids. They did come from several plants that were not clones. This information has been added in lines 113-115. Carrots of the “Jumbo” variety were bought from local producers at the end of the summer and stored at 4oC until use. 

L 120 – 121 What was the sample size of females in each temperature? I see that this information is shown in Figure 1, but I would like to see it in the text was well

Sample size added lines 133-134

L. 121 – 122 How many observations were performed?

5 observations were performed both at 10oC and at 40oC. Information added lines 120 and 121. 

L 123 – 124 Were these weevil eggs from a single female?

No, it was from a mixture of different females from the rearing. Information added lines 122-123. 

L 125 – 127 Is it possible to provide a figure of this experiment? (maybe as online resource)

A figure named ‘Experimental_setup’ has been added to Supplementaty_material. The reference to the figure has been added line 127.

Video analyzing

L 136 – 138 It is not clear to me why you only analyzed females that laid female progeny

Female that only laid sons were considered unmated. Because the behaviour of virgin females can differ from that of mated females, virgin females have been removed from the analyses. See lines 138-139. 

Offspring production

Ok

Data analyses

I am not familiar with the A Cox’s proportional hazards model, so I will assume authors made it right

Results

Looking at figure 2a I do not see much difference among classes, but your analyses show significant results. Have you used the correct statistical tests?

I double checked the statistical analyses and came out with the same results. I agree that differences in the proportion of each behaviour in relation to temperature are not obvious on the figure at first glance. 

Error-bars have been added to Figure 2a to make differences between means more visible. 

I suggest double checking the tests used. I found a Mann Whitney that is not described in the Methods.

A reference to Mann-Whitney test has been added in the Method in lines 171-172. Further details have been added to the statistical methods in lines 169-170 and 172-174. 

Discussion

All the topics shown in the Results are discussed. Nonetheless, the discussion is too long, more than 2000 words. It is difficult to keep the attention and focus with such a long discussion. Authors should speculate less and concentrate on their own results. A shorter and more objective discussion will be more valuable.

Following the request made by reviewer#2, we have reduced the length of the Discussion by removing several sentences: lines 287-289, 290-291, 289-290, 304-305, 333-335, 354. One paragraph has also been removed - lines 314-326. 

Figures

Very good! Mines are low resolution, but I think that the final version will have figures with high resolution.

High resolution figures will be uploaded with the revised version of the manuscript

---

## [Decision Letter · Decision Letter 1]

5 Jul 2021

The effect of temperature on host patch exploitation by an egg parasitoid

PONE-D-21-08181R1

Dear Dr. Julie Augustin,

We’re pleased to inform you that your manuscript has been judged scientifically suitable for publication and will be formally accepted for publication once it meets all outstanding technical requirements.

Kind regards,

Kleber Del-Claro, PhD

Academic Editor

PLOS ONE

Additional Editor Comments (optional): Congratulations for the excellent study!

Reviewers' comments:

Reviewer's Responses to Questions

**Comments to the Author**

1. If the authors have adequately addressed your comments raised in a previous round of review and you feel that this manuscript is now acceptable for publication, you may indicate that here to bypass the “Comments to the Author” section, enter your conflict of interest statement in the “Confidential to Editor” section, and submit your "Accept" recommendation.

Reviewer #1: All comments have been addressed

Reviewer #2: All comments have been addressed

2. Is the manuscript technically sound, and do the data support the conclusions?

Reviewer #1: Yes

Reviewer #2: Yes

3. Has the statistical analysis been performed appropriately and rigorously? 

Reviewer #1: Yes

Reviewer #2: Yes

4. Have the authors made all data underlying the findings in their manuscript fully available?

Reviewer #1: Yes

Reviewer #2: No

5. Is the manuscript presented in an intelligible fashion and written in standard English?

Reviewer #1: Yes

Reviewer #2: (No Response)

6. Review Comments to the Author

Reviewer #1: (No Response)

Reviewer #2: No comments. All suggestions were followed. I could not see the data being avaliable; I found neither a link not a sheet of it. But this concerns more the journal, as they require that data be available.

7. PLOS authors have the option to publish the peer review history of their article (what does this mean?). If published, this will include your full peer review and any attached files.

Reviewer #1: **Yes: **Bruno de Sousa-Lopes

Reviewer #2: No

---

## [Editor Report · Acceptance letter]

9 Jul 2021

PONE-D-21-08181R1 

The effect of temperature on host patch exploitation by an egg parasitoid 

Dear Dr. Augustin:

I'm pleased to inform you that your manuscript has been deemed suitable for publication in PLOS ONE. Congratulations! Your manuscript is now with our production department. 

Kind regards, 

on behalf of

Dr. Kleber Del-Claro 

Academic Editor

PLOS ONE